# Importance of Circadian Rhythms in the Ocular Surface

**DOI:** 10.3390/biom14070796

**Published:** 2024-07-04

**Authors:** Xiaozhao Zhang, Ying Jie

**Affiliations:** Beijing Key Laboratory of Ophthalmology and Visual Sciences, Beijing Institute of Ophthalmology, Beijing Tongren Eye Center, Beijing Tongren Hospital, Capital Medical University, No. 1 Dong Jiao Min Xiang, Dong Cheng District, Beijing 100730, China; zxz5360@mail.ccmu.edu.cn

**Keywords:** circadian, ocular surface, rhythms, clock

## Abstract

Circadian rhythms are a ubiquitous feature throughout the organism. Accumulating evidence suggests that the dysfunction of circadian rhythms due to genetic mutations or environmental factors contributes to the genesis and progress of multiple diseases. The physiological homeostasis of the ocular surface, like any other tissue or organ, is also orchestrated by circadian rhythms. In this review, we summarize the molecular clocks and the expression of clock-controlled genes in the mammalian ocular surface. Based on the circadian expression of these genes, we conclude the diurnal oscillations of cellular biological activities in the mammalian ocular surface. Moreover, we evaluate the factors entraining circadian oscillators in the ocular surface. Finally, we further discuss the latest development of the close correlation between circadian rhythms and ocular health. Briefly, this review aimed to synthesize the previous studies to aid in understanding the importance of circadian rhythms in the ocular surface and the possible opportunities for circadian rhythm-based interventional strategies to restore the homeostasis of the ocular surface.

## 1. Introduction 

Circadian rhythms are endogenous autonomous oscillators of biological activities shaped by the Earth’s rotation around its axis. This evolutionarily conserved timekeeping mechanism allows organisms to synchronize the internal processes with the environmental timing cues, ensuring optimal organismal adaptation [1]. The absence of the circadian clock system would hinder Homo sapiens from optimizing energy expenditure and the internal physiology of the body [2]. Studies in rodents have pinpointed the suprachiasmatic nucleus (SCN) as the anatomical structure of the central circadian clock [3]. The subsequent discoveries show that clock genes are universally expressed across cells in the body, which suggests that all tissues and cells potentially contain their own clocks [4].

Ocular tissues are proven to contain the core molecular circadian system and exhibit integrated rhythms of circadian activities [5,6,7]. The anatomical and physiological features of the eye undergo steadily diurnal oscillations [8]. The mammalian cornea shows strong circadian rhythms ex vivo, as shown by real-time analyses of the circadian expressions of m*Per2*: LUC [9,10,11]. Immune cells from the circulatory system are time-dependently recruited to the limbal vascular network. Additionally, the shape and thickness of human corneas vary diurnally [12]. Notably, it has been proven that disruptions of the normal light–darkness cycles are an independent risk factor for corneal dysplasia [13]. Continuous exposure to light can destroy the eye development of young chicks, leading to severe corneal flattening, corneal thickening, a shallow anterior chamber, and progressive hyperopia [14,15]. Moreover, renewal of the corneal epithelium has also been shown to follow a classical circadian rhythm [16,17,18]. The circadian variation affects the corneal epithelial wound-healing process, and the regenerative repair of the cornea by drugs targeting the clock loop can affect the re-epithelialization process of the cornea [19]. 

The circadian regulation of ocular surface functions pervades all vertebrate classes. However, since significant differences exist in the circadian organization of the cornea among these different systematic categories—and often among the different species within the same class—we decided to focus the review on mammals. This review presents general data about the molecular clock and its ocular surface target genes, while also updating the understanding of clock involvement in the regulation of ocular surface physiology and health.

## 2. The Molecular Clock and Clock-Driven Gene Expression in the Mammalian Ocular Surface

### 2.1. The Molecular Clock

The central circadian clock and the most peripheral clocks throughout the organism share a similar transcriptional architecture which is controlled by conserved, cell-autonomous, and self-sustaining mechanisms [20]. This molecular timekeeping system is generated by interlocking transcriptional translational feedback loops (TTFL; Figure 1) [21]. The activators circadian locomotor output cycles protein kaput (CLOCK) and brain and muscle ARNT-like 1 (BMAL1) heterodimerize to recognize E-box motifs, subsequently regulating the expressions of thousands of genes, named clock-controlled genes (CCGs) [22]. Among these, the repressors periods (PERs) and cryptochromes (CRYs) are translated and assembled into repressor complexes that inhibit the transcriptional activation function of the CLOCK-BMAL1 heterodimers. The activity and proteasomal degradation of repressor complexes are tightly controlled by posttranslational modifications, and their clearance alleviates CLOCK-BMAL1 inhibition for the start of another cycle [23]. The interlocking of this primary loop with additional activator–repressor pairs, for example, nuclear receptor subfamily 1 group D (REV-ERB) and RAR-related orphan receptor (ROR), allows for differentially phased cycles of transcription according to the distribution of dedicated motifs, including E-box, D-box, and ROR elements, across the genome [24]. These interconnected auto-regulatory loops, which exhibit an inherent 24 h periodicity, rely on nucleosome dynamics, long-range promoter–enhancer interactions, and a variety of co-factors and chromatin remodelers [24,25].

### 2.2. Clock-Driven Gene Expression in the Mammalian Ocular Surface

To dissect the effect of the endogenous circadian clock system on the genomic transcription of ocular surface tissue, Jiao, et al. analyzed corneal whole-gene transcriptomes by performing time-course genome-wide RNA sequencing (RNA-Seq) [26]. In the study, they collected the corneal tissue every 3 h over a 12:12 LD (light/dark) cycle (ZT3, 6, 9, 12, 15, 18, 21, and 24; ZT: zeitgeber time) from the adult male mice. The RNA-Seq results showed that 6034 (24.79%) of the genes in the cornea have a statistically significant rhythmic expression over an LD cycle. Most of the rhythmic genes in the cornea oscillate with a 24 h period. Further analysis showed that the number of the rhythmic genes with the peak distributed during the dark phase (ZT12-24) is more numerous than those in the light phase (ZT0-12). This is consistent with the physiological behavior of mice as nocturnal animals, characterized by being active at night and sleeping during the day [27]. In addition, the peak of the rhythmic gene distribution occurs at ZT12 during the diurnal transition [26]. 

## 3. The Diurnal Oscillations in the Mammalian Ocular Surface

The emergence of transcriptomic approaches in the field paved the way for a systematic analysis of temporal gene expression that functions in a specific tissue or cell type. Using this technique, it was demonstrated that thousands of transcripts exhibit circadian oscillations in various mouse organs, with approximately 43% of protein encoding genes displaying tissue-specific rhythmicity [26]. Recently, the first exhaustive analysis of rhythmic transcriptional expression profiles in >60 tissues/organs from a diurnal nonhuman primate was performed. This genome-wide transcriptome study uncovered that a wide array (about 82%) of protein-encoding genes, including both ubiquitous and tissue-specific ones, undergo rhythmic daily fluctuations. Most of the rhythm genes in the cornea oscillate with a 24 h period. GO analysis and KEGG enrichment analysis for these cycling genes show that the main pathways and biological processes of these genes are related to the basic activities of corneal cells, such as growth, proliferation, metabolism, and immunity. This conclusion is consistent with a recent study on the diurnal transcriptome of a primate and the nocturnal transcriptome of mice across major neural and peripheral tissues [28]. In addition, the RNA-Seq results of bulbar conjunctival swab samples collected from healthy subjects showed that the majority of rhythmically expressed genes were upregulated in the morning, which were involved in defense, cell turnover, and the regulation of gene expression, while the genes upregulated in the evening were involved in signaling and mucin production [29].

In addition to the 24 h rhythm found in mammals, it has recently been discovered that many shorter cycles of biological rhythms exist in mammals and other organisms [30,31,32]. Consistent with these studies, Li et al. found genome-scale rhythms with different oscillation periods, such as 12, 15, 18, and 21 h, in the cornea [12]. These data suggest that, as with other forms of physical oscillations, the transcriptional expression of genes in the cornea may also have a compound superimposed oscillation consisting of different periods, phases, and amplitudes. Two recent studies have focused on the features of the 12 h autonomous rhythmic genes in the liver [33,34]. Functional pathway studies show that these rhythmic genes are associated with protein quality control and processing and cell energy supply-related proteins. Similarly, Li et al. have found that a 12 h autonomous rhythmic transcription also exists in the murine cornea [12]. However, an enrichment analysis of 12 h cycling genes in the cornea shows that the cycling genes are closely related to basic functional pathways, such as carbohydrate metabolism, glycan biosynthesis and metabolism, amino acid metabolism, lipid metabolism, and nucleotide metabolism. These differences in functional pathways may be related to the tissue specificity of the liver and the cornea—that is, their different physiological functions. Further exploration of the internal and external factors that interfere with the 12 h rhythm transcriptome changes in the cornea may raise new research questions. 

Although mice (*Mus domesticus*) are considered to be representative models for studying mammalian gene expression rhythms, it is important to note that the C57BL/6J mice used in the present studies are naturally melatonin-deficient [35,36]; lacking the ability to produce melatonin [37,38]. The hormone melatonin (MLT), produced by the pineal gland, is a potent regulator of circadian rhythms or 24 h cycles of behavior and biological processes in other mammals [39]. Circulating MLT may reach the corneal tissue and entrain the corneal circadian clock by binding and activating the melatonin receptors [6]. Some evidence shows that melatonin receptors are expressed strongly in the mouse corneal epithelium, especially the melatonin type 2 (MT2) receptor [11]. The administration of exogenous MLT and MT2 agonists can phase shift the circadian rhythmic activity of the cornea. In addition, given that C57BL/6J mice are nocturnal animals, the circadian rhythm in human cornea should be hypothesized to exhibit a phase opposite to the rhythm in mice due to humans’ diurnal nature. Finally, mice are generally fed ad libitum and have fragmented daytime sleep. Therefore, these datasets are limited in their biological reflection of most other mammalian species, especially human beings.

In humans, the tear cytokine levels also exhibit diurnal variations, which may pertain to the circadian rhythm of the immune system of ocular surface tissues. A study showed that tear cytokine levels were generally higher in the evening than mid-day. In this study, it was found that only IL-10 and IL-1β levels had significant inter-day variations, while EGF, CX3CL1/fractalkine, CXCL10/IP-10, and VEGF were consistently higher in the evening compared to the mid-day measurements with a good intra-subject reproducibility [40]. However, another study suggested that IL-1β, IL-6, IL-10, IL-12p70, and TNF-α slightly increased in the morning and the late evening, while IL-8 remained low throughout the day [41]. These differences may be related to the number of subjects, detection sensitivity, and different tear sample collection times. These results can be used to determine the biomarkers of health and disease of the ocular surface and to establish the optimum time of day for sampling.

In summary, like other peripheral tissues, the cornea expresses most clock genes and other cycling genes with different periods, phases, and amplitudes. Most of the rhythmically expressed genes oscillate in periods of 12 h or 24 h, mainly associated with basic metabolic process pathways. These findings indicate a promising future for further study of the physiological activities of the cornea and their associations with certain corneal diseases, especially when these rhythmic oscillations are interrupted.

## 4. The Factors of Entraining Circadian Oscillators in the Ocular Surface the Circadian Rhythm of the Ocular Surface Is Influenced by Diabetes

Diabetes affects corneal morphology, metabolism, and physiology and causes diabetic keratopathy [42]. More than 70% of diabetic patients develop diabetic keratopathy and show some morphological changes in the cornea [43], including epithelial defects, recurrent epithelial erosions, delayed wound-healing, and ulcers. However, the cellular and molecular mechanisms underlying these pathological changes are poorly understood. Baba et al. showed that diabetes influenced the rhythmic expression of five core clock genes (*Clock*, *Bmal1*, *Per2*, *Cry1*, and *Rev-erbα*), as well as mitosis, in the normal murine corneal epithelium [11,43]. Diabetes also increased the emigration fluctuation of neutrophils and γ δ T-cells to the limbal region (Figure 2). Furthermore, the early administration of insulin to diabetic animals partially restored the alterations in rhythmic activities in the murine cornea. 

Core clock genes are the essential elements of circadian rhythm. Through positive and negative feedback loops, the circadian clock regulates the physiological activities of almost all cells, tissues, and organs. Similar to other tissues, it is shown that the expression of five clock genes (*Clock*, *Bmal1*, *Per2*, *Cry1*, and *Rev-erbα*) in the normal mouse cornea exhibit a robust circadian fluctuation, which is consistent with several recent observations [11,43]. Diabetes alters the circadian rhythm of the corneal epithelium in mice, such that the expression of *Clock*, *Bmal1*, and *Per2* is downregulated, and the expression of *Cry1* and *Rev-erbα* is upregulated. However, the significance of the altered expression levels of these genes and the mechanisms associated with corneal pathological changes remain unclear. 

The present study also demonstrates that insulin treatment restores the rhythmic expression of *Per2*, *Cry1*, and *Rev-erbα*. These findings are consistent with the ability of insulin to induce a phase shift in the peripheral clock, thereby advancing the phase of the feeding-related clock to correct the impaired rhythm of *Per2* expression in the liver and to partially normalize the heat rhythm in rats with STZ-induced diabetes [44]. These findings also suggest that insulin is able to synchronize the altered corneal circadian clock in diabetic animals. However, it is indicated that insulin supplementation by intraperitoneal injection has no effect on the circadian expression of *Clock* and *Bmal1*. A possible explanation for this finding might be that insufficient time had elapsed for the insulin treatment to allow the complete restoration of the expression of these genes. 

The thickness of the normal corneal epithelium is largely maintained by continuous mitosis and basal-cell migration to the surface. Traditionally, it is hypothesized to be dependent on three factors, termed the XYZ hypothesis by Thoft and Friend, where X represents the proliferation and anterior migration of basal epithelial cells, Y represents basal-cell migration from the periphery to the center of the cornea, and Z represents the desquamation of epithelial cells from the surface [45]. Based on this hypothesis, corneal epithelial mitosis is a core factor that maintains corneal epithelial integrity and homeostasis. The present study provides evidence for diurnal variations in the corneal mitosis of normal mice, corroborating previous studies that corneal mitosis and DNA synthesis exhibit circadian oscillations. And these oscillations are associated with the changes in light during the day and night. Through the use of 3H-TdR to label corneal epithelial cells, it was mentioned in an early study that the DNA synthesis of corneal epithelial cells reaches its peak during the night period in mice, and that the 3H-TdR incorporation peak precedes the mitotic peak by 4–6 h throughout the circadian cycle, which indicates that this period is necessary for cells to travel from mid-S-phase to M-phase [46]. However, the diabetic condition attenuates the circadian rhythm of corneal epithelial mitosis. This decline in the ability of cell division may provide a new clue for why diabetic patients are prone to corneal epithelial lesions. In addition, circadian rhythms influence generative processes after wounding. Corneal wounding at different times of the day reveals that the regenerative response of the corneal epithelium to injury is time-dependent. Moreover, epithelial proliferation is significantly altered. Hence, the results of Zagon et al. provide another explanation for delayed corneal wound-healing after the development of diabetes. Insulin treatment was able to partially normalize the circadian rhythm of mitosis [47]. Therefore, given the diurnal changes in corneal epithelial cell mitosis in animals with insulin-dependent diabetes, the altered self-renewal speed could contribute to delayed corneal wound-healing in diabetes [48]. Earlier studies have shown that insulin treatment ameliorates impaired corneal re-epithelialization in diabetic rats, and that topical application of insulin in rats with type 1 diabetes mellitus normalizes corneal wound-healing [49]. 

Research has found that the overall corneal dendritic cell density in healthy individuals remains reasonably constant throughout the sleep/wake cycle. However, there is a trend toward an overnight increase in the relative number of mature cells [50]. As this process is fundamental for the generation of adaptive immunity, the rhythmic variation in mature corneal dendritic cells may be related to the rhythmic changes in adaptive immune activities. Furthermore, leukocyte migration to peripheral tissues under homeostasis is critical for surveilling and combating pathogens. Recent evidence has shown that the immune system is controlled by the circadian clock, with oscillations in the number and activity of immune cells in the blood and tissues over the course of 24 h [48,51]. Consistent with this observation, Lavker et al. found diurnal rhythms in immune cell-trafficking to the corneal limbal region [46]. However, diabetes altered the circadian rhythm of neutrophils’ migration to the limbus. The peak of neutrophil trafficking time was shifted in diabetic mice. More importantly, the peak was significantly higher than that in the normal control group. This finding is consistent with studies of diabetic retinopathy, which have shown an increased number of static leukocytes in STZ-induced diabetic rats and neutrophil priming. The latter results in higher levels of superoxide and cytokines. Neutrophils are the main immune cells in the early phase of corneal wound-repair, which are essential for controlling and eliminating microbial infections from wound areas via phagocytosis, degranulation, and the release of neutrophil elastase and reactive oxygen species. However, there are two sides to neutrophils. For example, aberrant neutrophil-migration to the wound delays corneal wound-healing. Thus, dysfunctional neutrophil-trafficking to the cornea induces an inflammatory state. These findings show that insulin administration at an early stage could reverse this abnormal inflammatory state [52].

γδ T-cells are a lineage of innate-like lymphocytes that connect the innate and adaptive immune inflammatory responses. On the ocular surface, γδ T-cells constitutively express IL-17, which acts as a chemotactic factor to attract peripheral blood myeloid cells, such as neutrophils and monocytes. Lavker et al. found that γδ T-cell recruitment to the corneal limbal region, including both epithelial and stromal layers, showed a circadian rhythm that was shifted and enhanced by diabetes [46,47,53]. The increase in γδ T-cell numbers may explain the inflammatory state in the diabetic cornea. In addition, systemic insulin administration partially normalized the level and pattern of leukocyte-trafficking to the cornea. Therefore, the lack of insulin brought more immune cells to the cornea by altering the circadian system. This inflammatory state may potentially contribute to the delay in corneal wound-healing in diabetes [54].

## 5. Normal Circadian Rhythms Are Important for Ocular Physiology and Health

Alterations in ambient environmental light cycles not only affect the rhythmic expression of clock genes, but also interfere with diurnal physiological functions in peripheral tissues. For example, exposure to LL and DD conditions can disrupt *Bmal1* and *Per2* expression and impair developmental neovascularization in zebrafish. When mice are subjected to continuous light or to conditions of 12 h jet lag, the circadian release of hematopoietic stem cells into the peripheral blood is markedly altered, which is modulated by the sympathetic nervous system via circadian-controlled noradrenaline secretion. Likewise, early studies in rats and Japanese quail, along with our present data, show that oscillations in corneal epithelial mitosis are significantly altered in a constant light environment. These phase changes under different lighting conditions may be due to the free-running of the clock or a change in the phase angle of mitosis. Additionally, the decrease in the number of mitotic cells under the constant condition (LL) may also be due to the visible light. Some evidence shows that photons of visible light can damage DNA, membranes, and intracellular organs such as mitochondria and affect cellular respiration by producing reactive oxygen species and destroying cytochromes. Whether visible light can inhibit the mitosis of corneal epithelial cells and its possible range of strength need to be further assessed. Based on these findings, we propose that when environmental lighting is switched from day to night, the mitotic rhythm and clock gene expression in the corneal epithelium can be adapted through ambient lighting adjustments. 

Several studies highlight the role of circadian rhythms in tissue regeneration. Genetic evidence shows that the pattern of wound-healing in the skin is altered in circadian clock gene deficient *Per1*/*Per2*^mut^ mice and *Bmal1*^−/−^ mice. The elimination of Period (clock-repressor) proteins results in fibroblast and keratinocyte hyper-proliferation, whereas the elimination of the *Bmal1* and *Clock*-activator protein results in decreased epidermal cell proliferation and highly disorganized tissue granulation. Stem-cell regeneration in fruit fly intestines and mouse skin tissues follows a circadian rhythm. The disruption of clock components leads to division arrhythmia, stem cell aging, and delays in the healing process. Xue, et al. found that interruption of normal 24 h rhythms by exposure to continuous light or a jet-lag model, even over a short time, leads to a significant reduction and pattern changes in corneal epithelial mitosis [18]. These results indicate that environmental 24 h cycles are essential to maintain a healthy cornea. Moreover, regardless of the time of injury, the majority of corneal repair occurs during the late phase of the light/dark cycle. In fact, all groups exhibited the highest level of cell divisions immediately before the onset of darkness. This suggests that corneal wound-healing occurs in a clock-controlled manner; however, further experiments will be required to determine the precise underlying mechanisms. 

Circadian rhythms and energy metabolism are closely interlinked. Studies have shown that high fructose intake in mice significantly reprogrammed the circadian transcriptomic profiles of the normal cornea based on temporal and spatial distributions of epithelial mitosis, and immune cell trafficking, alongside pathways implicated in metabolism, neuronal activity, and immunity [55]. Similar results were also observed in the murine extraorbital lacrimal glands, which leads to a substantial remodeling of the preexisting rhythmic genes and the daily transcriptomic oscillation [56]. Likewise, excessive sucrose attenuates the oscillatory amplitude of *Per1/2* expression in rat small intestines, concurrently elevating the gene expression of GLUT5 and gluconeogenic enzymes [57]. These results indicate that daily nutrient intake can affect the expression of rhythm genes in peripheral tissues such as the ocular surface.

The time of day at which an injury occurs also affects the healing process. For example, when a partial hepatectomy is performed at different times of the day (ZT24 and ZT8), a distinct 8 h time lag in mitosis, from 40 to 48 h, is observed. The results from a kinetic analysis of zebrafish tailfin cell proliferation after truncation also showed that the healing rate depends on the time of day of the injury. After an amputation at ZT24, the number of dividing cells 10 h later was significantly greater than when amputation was performed at ZT12. Similarly, the number of epithelial cells undergoing mitosis after corneal injury in rats induced at noon (ZT6) was significantly higher than the number observed after a corneal injury induced at midnight (ZT18). This new knowledge may provide opportunities for circadian-targeted strategies to rapidly restore vision and shorten the time window of possible infection, ranging from the selection of corneal surgery timing to pharmacologic targeting of the molecular clock circuitry. Further studies will be aimed at deciphering the cellular mechanisms and signaling circuits within the corneal epithelium to determine how the circadian clock influences mitosis and wound-healing, possibly offering opportunities to design strategies for alternative therapies to accelerate healing. Furthermore, it will be interesting to investigate why such regulation has evolved (i.e., why it might be beneficial for mitosis and wound repair in the corneal epithelium to vary over a 24 h circadian cycle), or whether these changes are secondary consequences of circadian rhythms.

## 6. Conclusions and Future Prospects

The circadian clock system plays vital roles in the regulation of physiological processes in the ocular surface, including cell cycle progression, cytokine release, and immune function. Dry eye is proven to be closely associated with circadian rhythm disorders, so chronobiology research is becoming a focal point in the biological and medical fields at present. With the development of chronobiology in these years, external cues that can be used as synchronizers to reset the circadian oscillators of animals or cells have been gradually discovered. From time-serial collection to real-time monitoring through luciferase reporter genes and fluorescent proteins, the methods to observe circadian rhythms are rapidly advancing and becoming more diverse. In addition, researchers have also established a number of circadian clock-related databases to facilitate access to previous research results. Through the combination of various in vivo and in vitro experiments, the mechanisms underlying circadian oscillations are constantly being elucidated, and the complicated connections between circadian rhythm disorders and dry eye are also being identified. Illuminating the crosstalk between the circadian rhythm and ocular physiology can help us to better clarify the pathogenesis of dry eye, which provides new strategies and ideas for disease prevention and treatment.

## Figures and Tables

**Figure 1 biomolecules-14-00796-f001:**
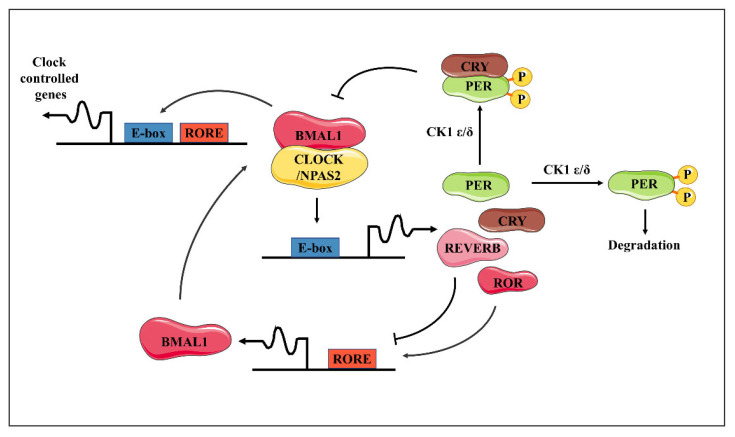
The molecular clock pathways and cornea clocks. Schematic representation of the transcriptional/translational feedback loop model for the molecular clock. The BMAL1/CLOCK (or BMAL1/NPAS2) dimer activates transcription of the *Per* and *Cry* genes upon binding to the E-box sequences in their promoters. In turn, the PER and CRY proteins form heterodimers able to inhibit the transcriptional activity of BMAL1/CLOCK, thus turning down their own transcription. Meanwhile, these factors undergo posttranslational modifications, in particular, the phosphorylation of PER proteins by the Casein Kinases 1d or 1e, signaling for ubiquitination and proteasomal degradation, and then allowing the cycle to restart. BMAL1/CLOCK likewise activates the expression of the *Rev-Erb* and *Ror* genes—the products of which, respectively, repress and activate transcription of the *Bmal1* gene at retinoic acid-related orphan receptor binding elements (RORE) sites. This generates an additional loop interlocked with the previous one, altogether contributing to the robustness of the clockwork. The presence of E-box and/or RORE sequences throughout the genome supports the rhythmic regulation of a set of target genes (CCG) for BMAL1/CLOCK, BMAL1/NPAS2, REV-ERB, and ROR transcription factors.

**Figure 2 biomolecules-14-00796-f002:**
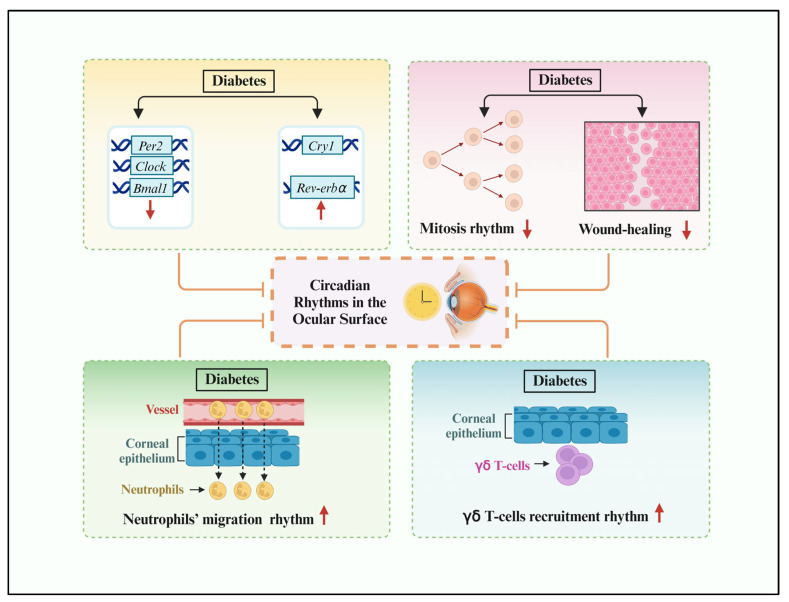
The circadian rhythm of the ocular surface is influenced by diabetes. Diabetes influences the rhythmic expression of five core clock genes (*Clock*, *Bmal1*, *Per2*, *Cry1*, and *Rev-erbα*) in the normal murine corneal epithelium. Diabetes attenuated the circadian rhythm of corneal epithelial mitosis with delayed corneal wound-healing. In addition, diabetes also enhanced the emigration fluctuation of neutrophils and the circadian rhythm of γ δ T-cells’ recruitment to the limbal region.

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
