# Peer review of "Importance of Circadian Rhythms in the Ocular Surface"

_biomolecules, 2024, doi:10.3390/biom14070796_

Round 1
Reviewer 1 Report
Comments and Suggestions for Authors
This is an interesting review on the expression of the molecular machinery that regulates circadian rhythms in the mammalian ocular surface, and their alterations mainly associated with diabetes. Specifically, there exist few studies on this field and this work summarized the existing studies in the field.
ISSUES:
1. It would be important that authors include the effect of nutrient intake on the circadian rhythm in cornea. There exist some studies that suggest that high fructose intake reprograms transcriptomic profiles in the normal mouse cornea, affecting spatial distribution of dividing cells, immune cells movement and signaling pathways (He et al., 2021, Invest Ophthalmol Vis Sci. 62:22.). Such issues should have important repercussions on metabolic health, mainly related with daily diet, as also occurs in rat small intestine (Sun et al., 2019, Chronobiol Int. 36:826-837), and in murine lacrimal glands (Lu et al., 2019, Invest Ophthalmol Vis Sci. 60:2038-2048).
2. It would be also interesting that authors include those results that describe the variation of mature dendritic cells in cornea, which increase overnight, while total dendritic cell density show relatively constant numbers along the day/night cycle (Alotaibi et al., 2022, Curr Eye Res 47:1239-1245), and perhaps comment about their meaning in relation to adaptive immunity.
3. There is an old paper that described the existence of circadian variations in limbal and corneal epithelium, showing a phase shift between labeling index of corneal epithelium with DNA precursors, and the mitotic index in such tissue, and shows a peak of DNA synthesis during darkness periods (Lavker et al., 1991, Invest Ophthalmol Vis Sci 32:1864-1875). Do authors believe necessary to mention this paper?
4. Please correct mistyping errors.
Author Response
Reviewer #1 Comments
- It would be important that authors include the effect of nutrient intake on the circadian rhythm in cornea. There exist some studies that suggest that high fructose intake reprograms transcriptomic profiles in the normal mouse cornea, affecting spatial distribution of dividing cells, immune cells movement and signaling pathways (He et al., 2021, Invest Ophthalmol Vis Sci. 62:22.). Such issues should have important repercussions on metabolic health, mainly related with daily diet, as also occurs in rat small intestine (Sun et al., 2019, Chronobiol Int. 36:826-837), and in murine lacrimal glands (Lu et al., 2019, Invest Ophthalmol Vis Sci. 60:2038-2048).
Response: Thanks very much for the reviewer’s professional comments. We have included a description regarding the effect of nutrient intake on the circadian rhythm in cornea, as suggested. In the revised manuscript, we have added this paragraph: “Circadian rhythms and energy metabolism are closely interlinked. Studies have shown that high fructose intake in mice significantly reprogrammed the circadian transcriptomic profiles of the normal cornea based on temporal and spatial distribu-tions of epithelial mitosis, and immune cell trafficking, alongside pathways implicated in metabolism, neuronal activity, and immunity. The similar results were also observed in the murine extraorbital lacrimal glands, which leads to a substantial remodeling of the preexisting rhythmic genes and the daily transcriptomic oscillation. Likewise, excessive sucrose attenuates the oscillatory amplitude of Per1/2 expression in rat small intestines, concurrently elevating the gene expression of GLUT5 and gluconeogenic enzymes. These results indicate that daily nutrient intake can affect the expression of rhythm genes in periph-eral tissues such as the ocular surface.” (Please see Lines 311-321)
- It would be also interesting that authors include those results that describe the variation of mature dendritic cells in cornea, which increase overnight, while total dendritic cell density show relatively constant numbers along the day/night cycle (Alotaibi et al., 2022, Curr Eye Res 47:1239-1245), and perhaps comment about their meaning in relation to adaptive immunity.
Response: We appreciate the reviewer for the professional comments. We have included a description of the variation of dendritic cell density in cornea, as per the reviewer’s recommendation. In the revised manuscript, we have added the sentences “Research has found that the overall corneal dendritic cell density in healthy indi-viduals remains reasonably constant throughout the sleep/wake cycle. However, there is a trend toward an overnight increase in the relative number of mature cells. As this process is fundamental for the genera-tion of adaptive immunity, the rhythmic variation of mature corneal dendritic cells may be related to the rhythmic changes in adaptive immune activities.” (Please see Lines 238-243)
- There is an old paper that described the existence of circadian variations in limbal and corneal epithelium, showing a phase shift between labeling index of corneal epithelium with DNA precursors, and the mitotic index in such tissue, and shows a peak of DNA synthesis during darkness periods (Lavker et al., 1991, Invest Ophthalmol Vis Sci 32:1864-1875). Do authors believe necessary to mention this paper?
Response: We appreciate the reviewer for the constructive comments. We think this paper is very suitable to be mentioned in the manuscript. So in the revised manuscript, we have added a description regarding the variation of the rhythmic proliferation of corneal epithelial cells: “Through the use of 3H-TdR to label corneal epithelial cells, it was mentioned in an early study that DNA synthesis of corneal epithelial cells reached its peak during the night period in mice, and that the 3H-TdR incorporation peak precedes the mitotic peak by 4-6 hours throughout the circadian cycle, which indicated that this period was neces-sary for cells to travel from mid S-phase to M-phase.” (Please see Line 219-224).
- Please correct mistyping errors.
Response: We appreciate the reviewer for pointing out this issue. In the revised manuscript, we have carefully checked the full text of the manuscript and corrected all kinds of spelling, grammar and formatting mistakes.
Reviewer 2 Report
Comments and Suggestions for Authors
The paper titled "Importance of Circadian Rhythms in the Ocular Surface" is a very useful and original well written review which is of interest for the readers. The introduction provides the necessary information for the further understanding of the topic, the paper is written in a scientifically sound style and the work is well organized. The description of the molecular clock and clock-driven gene expression in the mammalian ocular surface is followed by the presentation of the diurnal oscillations in the mammalian ocular surface. Further on, the authors illustrate the factors of entraining circadian oscillators in the ocular surface and show that the circadian rhythm of the ocular surface is Influenced by diabetes. The authors also describe the manner in which normal circadian rhythms are important for ocular physiology and health. Finally, the authors state that establishing the connection between circadian rhythm and ocular physiology is bringing a contribution in deciphering the pathogenesis of the dry eye with the aim to identify new strategies for the prevention and treatment of dry eye.
The paper meets the criteria for being published in "Biomolecules"
Author Response
Reviewer #2 Comments
Reviewer #2: The paper titled "Importance of Circadian Rhythms in the Ocular Surface" is a very useful and original well written review which is of interest for the readers. The introduction provides the necessary information for the further understanding of the topic, the paper is written in a scientifically sound style and the work is well organized. The description of the molecular clock and clock-driven gene expression in the mammalian ocular surface is followed by the presentation of the diurnal oscillations in the mammalian ocular surface. Further on, the authors illustrate the factors of entraining circadian oscillators in the ocular surface and show that the circadian rhythm of the ocular surface is Influenced by diabetes. The authors also describe the manner in which normal circadian rhythms are important for ocular physiology and health. Finally, the authors state that establishing the connection between circadian rhythm and ocular physiology is bringing a contribution in deciphering the pathogenesis of the dry eye with the aim to identify new strategies for the prevention and treatment of dry eye.
The paper meets the criteria for being published in "Biomolecules"
Response: We really appreciate the reviewer for the professional comments on this manuscript, and it is a great honor for us to be recognized.
Round 2
Reviewer 1 Report
Comments and Suggestions for Authors
Manuscript has been improved by authors after considering previous observations. However, they should take into account the following issues:
1.- Since manuscript describes the importance of circadian rhythms in ocular surface, authors should cite work carried out in conjunctiva (Alenezi et al., 2022, Differential gene expression of the healthy conjunctiva during the day. Cont Lens Anterior Eye. 45(4):101494. doi: 10.1016/j.clae.2021.101494. ).
2.-Moreover, the effect of high fructose has been described directly in cornea (He et al., 2021, Short-Term High Fructose Intake Impairs Diurnal Oscillations in the Murine Cornea. Invest Ophthalmol Vis Sci. 62(10):22. doi: 10.1167/iovs.62.10.22.) and therefore, should be cited.
3.- They also should consider to discuss the diurnal variation o cytokine levels in tears (Benito et al., 2014, Intra- and inter-day variation of cytokines and chemokines in tears of healthy subjects. Exp Eye Res. 120:43-49. doi: 10.1016/j.exer.2013.12.017; Uchino et al., 2006, Alteration pattern of tear cytokines during the course of a day: diurnal rhythm analyzed by multicytokine assay. Cytokine. 33(1):36-40. doi: 10.1016/j.cyto.2005.11.013), because such variation should modify the behavior of the ocular tissues.
Author Response
Reviewer Comments
- Since manuscript describes the importance of circadian rhythms in ocular surface, authors should cite work carried out in conjunctiva (Alenezi et al., 2022, Differential gene expression of the healthy conjunctiva during the day. Cont Lens Anterior Eye. 45(4):101494. doi: 10.1016/j.clae.2021.101494. ).
Response: Thanks very much for the reviewer’s professional comments. We have included a description regarding the circadian rhythms of conjunctiva, as suggested. In the revised manuscript, we have added these sentences: “Besides, the RNA-Seq results of bulbar conjunctival swab samples collected from healthy subjects showed that the majority of rhythmically expressed genes were upreg-ulated in the morning, which were involved in defense, cell turnover and regulation of gene expression, while the genes upregulated in the evening were involved in signaling and mucin production.” (Please see Lines 126-130)
- Moreover, the effect of high fructose has been described directly in cornea (He et al., 2021, Short-Term High Fructose Intake Impairs Diurnal Oscillations in the Murine Cornea. Invest Ophthalmol Vis Sci. 62(10):22. doi: 10.1167/iovs.62.10.22.) and therefore, should be cited.
Response: We appreciate the reviewer for pointing out the issue. We apologize for inserting the wrong reference, which should be the 52nd citation in the original manuscript (now the 55th citation). In the revised manuscript, we have corrected it into the citation (He et al., 2021, Short-Term High Fructose Intake Impairs Diurnal Oscillations in the Murine Cornea. Invest Ophthalmol Vis Sci. 62(10):22. doi: 10.1167/iovs.62.10.22.) (Please see Line 337).
- They also should consider to discuss the diurnal variation o cytokine levels in tears (Benito et al., 2014, Intra- and inter-day variation of cytokines and chemokines in tears of healthy subjects. Exp Eye Res. 120:43-49. doi: 10.1016/j.exer.2013.12.017; Uchino et al., 2006, Alteration pattern of tear cytokines during the course of a day: diurnal rhythm analyzed by multicytokine assay. Cytokine. 33(1):36-40. doi: 10.1016/j.cyto.2005.11.013), because such variation should modify the behavior of the ocular tissues.
Response: We appreciate the reviewer for the constructive comments. We have included a description regarding the diurnal variation of cytokine levels in tears, as suggested. In the revised manuscript, we have added the paragraph: “In humans, the tear cytokine levels also exhibit diurnal variations, which may pertain to the circadian rhythm of the immune system of ocular surface tissues. A study showed that tear cytokine levels were generally higher in the evening than in the mid-day. In this study, it was found that only IL-10 and IL-1β levels had significant in-ter-day variations, while EGF, CX3CL1/fractalkine, CXCL10/IP-10, and VEGF were con-sistently higher in the evening compared to the mid-day measurements with good in-tra-subject reproducibility. However, another study sug-gested that IL-1β, IL-6, IL-10, IL-12p70, and TNF-α slightly increased in the morning and the late evening, while IL-8 remained low throughout the day. These differences may be related to the number of sub-jects, detection sensitivity and different tear sample collection times. These results can be used to determine the biomarkers of health and disease of the ocular surface and to establish the optimum time of day for sampling.” (Please see Line 166-177).